# The Strategy and Application of Gene Attenuation in Metabolic Engineering

**DOI:** 10.3390/microorganisms13040927

**Published:** 2025-04-17

**Authors:** Yahui Zhang, Zhaoxia Jin, Linxia Liu, Dawei Zhang

**Affiliations:** 1School of Biological Engineering, Dalian Polytechnic University, Dalian 116034, China; zhangyahui@tib.cas.cn; 2Tianjin Institute of Industrial Biotechnology, Chinese Academy of Sciences, Tianjin 300308, China; zhang_dw@tib.cas.cn; 3University of Chinese Academy of Sciences, Beijing 100049, China; 4State Key Laboratory of Engineering Biology for Low-Carbon Manufacturing, Tianjin Institute of Industrial Biotechnology, Chinese Academy of Sciences, Tianjin 300308, China

**Keywords:** gene attenuation, metabolic engineering, metabolic pathway, gene regulation

## Abstract

Metabolic engineering has a wide range of applications, spanning key sectors such as energy, pharmaceuticals, agriculture, chemicals, and environmental sustainability. Its core focus is on precisely modulating metabolic pathways to achieve efficient, sustainable, and environmentally friendly biomanufacturing processes, offering new possibilities for societal sustainable development. Gene attenuation is a critical technique within metabolic engineering, pivotal in optimizing metabolic fluxes and improving target metabolite yields. This review article discusses gene attenuation mechanisms, the applications across various biological systems, and implementation strategies. Additionally, we address potential future challenges and explore its potential to drive further advancements in the field.

## 1. Introduction

### 1.1. Overview of Metabolic Engineering and Its Significance

Metabolic engineering is the science of controlling cellular activities by manipulating enzymatic reactions, transport processes, and regulatory functions within the cell [1]. Metabolic engineering aims to appropriately reconfigure cellular metabolism to construct genetically engineered strains that serve as robust cellular factories for various purposes, including the biosynthesis of target compounds [2]. The metabolic pathways of microorganisms or cells can be optimized through metabolic engineering to improve yield and efficiency in production processes [3]. This is of substantial importance for the industrial production of pharmaceuticals, chemicals, biofuels, and other products [4]. Metabolic engineering typically employs recombinant DNA technology to direct targeted strain improvements, aiming to enhance the performance of biological processes, such as increasing the yield of metabolic products and protein outputs or altering the distribution and rate of metabolic pathways [1,5]. This necessitates the modification of genes encoding key enzymes or regulatory proteins within metabolic pathways, thereby enabling the modulation of metabolite flux across different routes, enhancing the production of the desired product, or improving cellular efficiency. Strategies include gene overexpression, gene knockout, or gene attenuation, enabling precise control over pathway efficiency. Gene overexpression involves increasing the expression level of a target gene by introducing additional copies or enhancing its expression [6]. This approach is commonly employed to study gene functions or to enhance specific cellular processes. Gene knockout entails the complete removal or inactivation of a target gene, with widely used techniques like CRISPR/Cas9 [7]. Knocking out a gene often leads to substantial functional changes at the cellular or organismal level [8,9]. Gene attenuation refers to the reduction in gene expression, which can be achieved through methods like CRISPR interference (CRISPRi) [10], small interfering RNA (siRNA) to suppress target gene transcription, or by modulating transcription factors for partial gene suppression.

### 1.2. Development of Gene Editing Technology

Gene editing technology is a technology that can modify the genome and its transcription products of organisms. Early gene editing technologies relied on the homologous recombination pathway within cells to insert exogenous DNA sequences into the genome [11]. By adding homologous arms to both ends of the exogenous DNA sequence, precise integration of the exogenous sequence could be achieved. However, because the DNA recognition and cleavage functions of homing endonucleases are located in the same domain, their editing sites are limited by the sequence. Therefore, researchers turned their attention to Zinc Finger Nucleases (ZFNs) [12,13,14] and transcription activator-like effectors [15,16].

The CRISPR/Cas system is currently the most widely used gene editing tool. In 1987, the Nakata research group at Osaka University in Japan first discovered five 29-nt-long repetitive palindromic sequences in the 3′-flanking region of the *iap* gene in *Escherichia coli* (*E. coli*). These sequences were separated by 32-nt non-repetitive sequences, and the five repetitive sequences were not homologous to any known prokaryotic sequences [17]. In 2000, the Mojica research group at the University of Alicante in Spain found this type of repetitive sequence in various prokaryotic organisms [18]. The CRISPR/Cas9 system has now been applied in a variety of fields, such as regulating gene expression in vivo [19], constructing animal disease models [20], studying gene regulatory networks within cells, and high-throughput screening of genes [21,22,23,24]. Meanwhile, researchers have conducted in-depth explorations of the CRISPR/Cas system. There are diverse types of Cas proteins in bacteria, each with its own characteristics, including PAM sequences, protein size, and cleavage activity, which has expanded the application range of the CRISPR/Cas system [25,26]. Due to its flexible mechanism of action, ease of operation, and diversity, the application and methodological research of CRISPR/Cas have developed rapidly. Gene attenuation is an important application direction of gene editing technology, which focuses on the fine regulation of gene function by regulating gene expression level.

### 1.3. The Importance of Gene Attenuation in Metabolic Engineering

Gene regulation plays a pivotal role, especially in metabolic engineering. As shown in Table 1, differences in gene regulation are critical. Among them, gene attenuation stands out compared to gene knockout and overexpression. By reducing, rather than eliminating, target gene expression, gene attenuation allows for precise control of enzyme activity within metabolic pathways. This is especially crucial at pathway nodes where balanced flux is needed. Full inhibition, as in knockout, can cause metabolic bottlenecks or unwanted byproduct accumulation, whereas attenuation allows for an optimized balance, enhancing target metabolite yield and avoiding negative effects. While gene overexpression can increase key enzyme levels, it may also impose a high metabolic burden by consuming excessive resources. In contrast, gene attenuation can moderately reduce competing metabolite synthesis, optimizing resource allocation and increasing target product yield. In synthesis pathways, certain competitive branches can deplete essential precursors. By knocking down genes for key enzymes in these branches, precursor losses can be decreased and redirected towards target product synthesis. In contrast, gene knockout could fully block branches, potentially disrupting metabolic balance and cell growth [27]. In complex metabolic networks, a single pathway gene knockout often triggers compensatory reactions in other pathways, while gene attenuation minimally disrupts the network, better preserving overall stability and cell viability. Thus, gene attenuation enables pathway optimization without critically impairing cell growth or activating non-target metabolic pathways. Consequently, gene attenuation provides more precise and flexible control than gene knockout or overexpression in metabolic engineering, offering unique advantages in improving target product yields, optimizing resource distribution, and maintaining cell health. This makes it an essential strategy in metabolic engineering [28,29].

### 1.4. Development and Mechanisms of Gene Attenuation Technology

Gene attenuation refers to the partial reduction or modulation of a gene’s expression or function, allowing the gene to retain some activity level while considerably lowering its overall function. This approach does not eliminate the gene’s function but instead “weakens” its effect. Antisense RNA (asRNA), small RNAs (sRNAs), RNA interference (RNAi), CRISPRi, regulatory promoters, and ribosome binding sites (RBS) can enable precise control of target gene expression levels. Transcription Activator-Like Effector Nucleases (TALENs) [34] and ZFNs [35] also contribute to gene expression regulation. Computational biology and modeling can facilitate simultaneous regulation of multiple genes in metabolic pathways to achieve flux optimization. At the same time, the design of regulatory sequences allows for fine-tuning of expression level (Figure 1). Commonly used hosts for gene attenuation include *E. coli*, *Clostridium acetobutylicum*, yeast, filamentous fungi, rice, ginseng, *Nicotiana tabacum*, mouse, cow, and mammalian cells. Computational biology and modeling are frequently applied to microorganisms, animals, and plants. As RNA technology is commonly used in prokaryotes, eukaryotes, and plants; RNAi and TALENs/ZFNs are applied in both animals and plants; SRNAs, promoter optimization, RBS optimization, and CRISPRi are used in both prokaryotes and eukaryotes.

The Jorgensen lab made a groundbreaking discovery when they introduced the anthocyanin synthesis-related gene, Chalcone Synthase (CHS), into the morning glory plant and observed an unexpected phenomenon: the flower color changed. This phenomenon was termed “co-suppression,” representing the interaction effect between endogenous and exogenous genes [36]. Kooter and Agrawal later discovered that this process involves the degradation of RNA post-transcription and thus redefined the phenomenon as “post-transcriptional gene silencing” (PTGS) [37,38]. Similarly, Cogoni and Sweykowsa-Kulinska observed transgene-induced gene silencing in fungi, and their research confirmed that DNA methylation is not a necessary condition for gene silencing [39,40]. Subsequent studies by Fire [41] and colleagues revealed the phenomenon of RNAi, a process in which double-stranded RNA (dsRNA) specifically triggers the degradation of homologous mRNA, thereby reducing or completely inhibiting gene expression [42].

AsRNA is a small, diffusible, and highly structured RNA molecule that interacts with target RNA (referred to as sense RNA) through sequence complementarity [43]. AsRNA was first identified in the ColE1 plasmid of *E. coli* [44]. Naturally occurring asRNAs are widespread in both prokaryotes and eukaryotes, where they regulate gene expression at multiple levels, including transcription, RNA editing, post-transcriptional regulation, and translation. AsRNA primarily exerts its regulatory functions by forming stable double-stranded structures with mRNA through complementary base pairing. These structures modulate gene expression through the following mechanisms: Inhibition of Translation: The double-stranded structure formed by asRNA and mRNA can block ribosome binding sites or hinder ribosomal progression along the mRNA, thereby reducing the expression of the target gene [5]. Promotion of mRNA Degradation: Certain asRNAs function similarly to RNAi. Upon pairing with mRNA, the asRNA-mRNA duplex can recruit RNA-induced silencing complexes (RISC) or nucleases, leading to targeted recognition and cleavage. Alteration of Protein Structure: Some asRNAs modulate gene expression by altering the splicing patterns of pre-mRNAs upon binding, thereby influencing the expression levels of the target gene. AsRNA can bind specifically to DNA sequences in the promoter region. Such interactions may disrupt the binding of essential transcriptional regulatory proteins, thereby reducing transcription initiation and weakening gene expression at the source. Modification of Chromatin Structure: Chromatin structure plays a crucial role in regulating gene expression. AsRNAs can recruit chromatin-modifying enzymes to chemically alter chromatin or directly interact with specific DNA sequences or histones. These interactions can modify the physical state of chromatin, influencing gene accessibility and expression levels [45].

Small regulatory RNA (sRNA) refers to a class of non-coding RNA molecules found in both prokaryotes and eukaryotes. These RNAs can influence the translation process and mRNA stability, thereby modulating the activity of specific genes [46]. In prokaryotes, sRNAs regulate gene expression by forming complementary base pairs with target mRNAs, affecting their stability or translation [47]. In eukaryotes, sRNAs encompass various types, such as microRNAs (miRNAs) and piwi-interacting RNAs (piRNAs), which also participate in post-transcriptional regulation [48]. sRNAs can be used independently to weaken gene expression or combined with genome-editing technologies, such as CRISPRi, for more precise gene regulation [49]. When integrated with the CRISPRi system, sRNAs can work in conjunction with the dCas9 protein to form an inhibitory complex at the promoter region or transcriptional start site of the target gene, thereby achieving transcriptional repression. This combination enhances the regulatory effects of sRNAs, allowing for more efficient and precise gene silencing.

RNAi is a post-transcriptional gene-disturbing mechanism that utilizes small interfering RNAs (siRNAs) or microRNAs (miRNAs) to degrade target mRNA or inhibit its translation [38]. SiRNA is the core element of RNAi. SiRNA sometimes referred to as short interfering RNA or silencing RNA, is a double-stranded RNA molecule 20 to 25 nucleotides in length. siRNAs are generated by the cleavage of double-stranded RNA (dsRNA) by the nuclease Dicer [50]. Each siRNA strand is approximately 20–25 nucleotides in length and can pair with target mRNAs, promoting their degradation via the RNA-induced silencing complex (RISC) [51]. Dicer enzyme: This nuclease specializes in processing long dsRNA into short siRNA. Dicer plays a pivotal role in RNAi by precisely cleaving dsRNA to produce duplex siRNAs. RISC is a protein complex containing siRNA and associated auxiliary proteins [52]. It can accurately recognize and bind to target mRNA, facilitating mRNA degradation via the complementary base pairing of siRNA with the mRNA [53]. RNA-dependent RNA polymerase (RdRP) is a specialized enzyme capable of synthesizing new RNA strands using siRNA as a template, a process known as RdRP. Following the production of siRNA by Dicer and its integration into RISC, target mRNA degradation is initiated. Subsequently, RdRP uses siRNA as a template to synthesize new RNA chains, amplifying the RNAi effect [54,55]. Based on this principle, researchers use a short dsRNA to induce efficient and specific degradation of homologous mRNA within the cell, thereby achieving the goal of suppressing target gene expression. Currently, two main approaches are used to induce RNAi in mammalian cells: one is the exogenous synthesis of siRNA, which is transfected into cells to induce transient silencing of the target gene; the other is the use of plasmids or viral vectors to express short hairpin RNA (shRNA) [56], resulting in sustained and stable induction of RNAi within the cells [57]. RNAi using the shRNA expression vector system has been successfully applied in various animal models, including mice [58], rats, guinea pigs, pigs [59], and embryos of sheep and cows [60].

The CRISPR-Cas system is a gene-editing tool derived from the adaptive immune mechanisms of bacteria and archaea. It consists of two main components: the CRISPR array, which is made up of short repetitive DNA sequences, and the associated Cas proteins. Based on their evolutionary classification, CRISPR-Cas systems are primarily divided into two main classes: Class 1 and Class 2. Class 1 systems are characterized by the involvement of multi-subunit crRNA effector complexes, while Class 2 systems rely on a single protein, such as Cas9, to perform all effector functions [61]. Within these two main classes, there are many subtypes defined by the presence and arrangement of specific signature genes. Different subtypes possess unique combinations of expression, interference, and adaptation modules, providing strong support and possibilities for diverse gene editing applications [59]. While traditional genetic engineering techniques support the coordinated manipulation of multiple genes within metabolic pathways, the CRISPR-Cas system offers an alternative approach that allows for the simultaneous suppression of multiple gene expressions, thereby circumventing the cumbersome process of targeting individual genes [60,61,62,63,64,65].

CRISPRi is a Type II CRISPR-Cas system that distinguishes itself from other CRISPR-Cas systems by using a dCas9 protein. This dCas9 protein is a variant of the Cas9 protein with its RuvC and HNH nuclease domains inactivated while retaining DNA-binding capability. With the help of single-guide RNA (sgRNA), it can specifically target desired sites [66]. The sgRNA molecule contains four structural components: a trc promoter, a 20-nucleotide region complementary to the target DNA, a 42-nucleotide dCas9-binding hairpin, and a 40-nucleotide transcription terminator derived from Streptococcus pyogenes. When co-expressed with dCas9, changing the sgRNA sequence allows dCas9 to bind to virtually any sequence in the genome. dCas9 then binds to the promoter or coding regions, blocking transcription by acting as a barrier for RNA polymerase. The precise target is defined by base pairing between the sgRNA and the protospacer adjacent motif (PAM) sequence [63]. Moreover, the CRISPR-Cas9 complex can also facilitate transcriptional activation [67]. One of the most significant aspects of this system is the precise control over sgRNA or dCas9 expression through induction, which is crucial when addressing issues like the basal expression levels of target genes or the accumulation of toxic substances [68].

A promoter is an upstream DNA sequence of a gene that includes key elements such as the core promoter, enhancer/silencer binding sites for transcription factor (TF) interaction, and regulatory elements that respond to environmental signals [69] (e.g., hormones or temperature). As a fundamental element of gene transcription and expression, promoters are widely employed in strategies for regulating gene expression. Beyond enhancing the strength of promoters associated with target product synthesis, modifying promoters to weaken gene expression is also a common approach for regulating metabolic pathways. Several strategies for reducing promoter activity through modifications include the following: Altering the consensus sequence: Modifications to the consensus sequence can reduce RNA polymerase affinity [70], thereby lowering transcriptional activity. This directly affects the efficiency of RNA polymerase binding to the promoter. Adjusting the spacer length between the −35 and −10 regions: Changes in the spacer length can influence the proper positioning and functionality of RNA polymerase. Increasing or decreasing this distance appropriately can slow down the binding or initiation efficiency of RNA polymerase, thereby modulating gene expression levels. Reducing or modifying transcription factor binding sites within the promoter: By decreasing or altering specific TF binding sites, the activation effect of certain transcription factors on gene expression can be diminished, achieving a reduction in gene expression. These strategies not only effectively lower promoter activity but also provide greater flexibility in metabolic regulation for industrial microorganisms, enabling them to meet specific production requirements.

The attenuation of gene expression via the ribosome binding site (RBS) is achieved by modulating the translation initiation process, reducing the efficiency of ribosome binding to mRNA, and thereby decreasing the production of the target protein. This regulatory strategy is extensively applied in synthetic biology and metabolic engineering for precise control of gene expression levels. The RBS is a short sequence located upstream of the start codon in mRNA, primarily responsible for recruiting ribosomes by enabling complementary interactions between the RBS and the 3′ end of the 16S rRNA within the ribosomal small subunit, thereby facilitating the formation of the translation initiation complex. Furthermore, the strength of the RBS sequence directly determines the frequency of ribosome binding, thus influencing the overall translation efficiency. In prokaryotes, the RBS typically corresponds to the Shine-Dalgarno sequence, positioned approximately 6–8 nucleotides upstream of the start codon, with its complementarity to the 16S rRNA being critical for translation initiation.

The attenuation of RBS-mediated translation efficiency can be achieved through various approaches. Modifications to the RBS sequence [71], such as point mutations or sequence optimization, can be employed to reduce its complementarity to the 16S rRNA, thereby diminishing ribosome binding efficiency. Synthetic biology techniques further enable the design of RBS libraries with varying strengths, allowing the identification of sequences that provide optimal attenuation effects. Adjusting the distance between the RBS and the start codon is another effective method, as this spacing critically influences translation efficiency; deviations from the optimal range, whether through elongation or shortening, can impair ribosome binding and reduce translation initiation. Additionally, physical barriers, such as secondary structures (e.g., stem-loops) or repetitive sequences, can be introduced upstream of the RBS to hinder ribosome access. Dynamic regulation can also be achieved through the use of sRNAs or other regulatory elements that competitively bind to the RBS, thereby obstructing ribosome binding in a controllable and responsive manner. These strategies collectively provide a versatile toolkit for the fine-tuning of protein expression in diverse biological systems. These strategies enable effective attenuation of gene expression and provide flexible tools for metabolic regulation in various applications.

TALENs and ZFNs are gene-editing tools that enable precise modifications to the promoter or coding regions of genes, thereby regulating gene expression [72]. ZFNs originated from the discovery of zinc finger proteins in 1984 in the transcription factors of *Xenopus laevis*. Following artificial engineering and fusion with nucleases, the technology evolved into ZFNs. A ZFN consists of two key components: zinc finger proteins (ZFPs) and the FokI endonuclease [34]. ZFPs, composed of tandem arrays of zinc finger motifs, recognize and bind specific DNA sequences, while FokI endonuclease mediates sequence-specific DNA cleavage by forming a dimeric complex [73]. The DNA-binding domain of ZFPs contains multiple Cys2-His2 zinc finger motifs, with each motif typically recognizing three specific base pairs. Tandemly arranged zinc finger domains confer high specificity to the ZFP for its target sequence [74,75]. The FokI nuclease, derived from *Flavobacterium okeanokoites*, is a non-specific cleavage domain located at the C-terminus, composed of 96 amino acids. FokI requires dimerization for enzymatic activity. When two ZFNs bind to adjacent recognition sites separated by 6–8 base pairs, the FokI nuclease domains dimerize, enabling the precise cleavage of double-stranded DNA.

TALENs, which emerged as an alternative to ZFNs, were developed following the discovery of transcription activator-like effectors (TALEs) in 2007 from *Xanthomonas* bacteria. TALEs are secreted proteins capable of binding host plant genomes and activating transcription. By 2012, TALEN technology was introduced [18]. A TALEN consists of two components: the DNA recognition domain, derived from TALEs, and the same type IIS FokI nuclease used in ZFNs [76]. The TALE domain binds to specific DNA sequences with high specificity, while the FokI nuclease mediates double-strand breaks (DSBs) via dimerization. Following cleavage, endogenous cellular repair pathways, such as non-homologous end joining (NHEJ) or homology-directed repair (HDR), are employed to repair the induced DSBs. This technology has since largely replaced ZFNs due to its greater simplicity and modularity in targeting specific DNA sequences.

In the study and application of gene attenuation, computational biology and modeling play a critical role. These tools not only facilitate the understanding of the complex dynamics of gene regulatory networks but also provide theoretical support and precise predictions for the design and optimization of gene attenuation strategies.

Systems biology modeling enables the dissection of gene regulatory networks. Dynamic system modeling, employing differential equations or discrete frameworks, is used to mathematically characterize the transcriptional, translational, and metabolic behavior of target genes. For instance, by examining the impact of RBS attenuation on translation efficiency, such models can predict dynamic changes in translation products. Steady-state analysis and sensitivity analysis further allow the computation of gene expression levels under equilibrium conditions and the evaluation of various regulatory strategies (e.g., RBS mutations or CRISPRi) to identify the most influential nodes in the network.

Data-driven approaches enable prediction and optimization in gene attenuation, as well as the precise identification of targets for attenuation. Firstly, a gene expression prediction model based on deep learning is proposed, which can infer the expression level from the gene sequence. This method can be used to predict the translation efficiency of RBS sequences and provide support for gene-weakening design [77]. Integration of transcriptomics, proteomics, and metabolomics data provides insights into the global impact of gene attenuation on biological systems, supporting the design of more effective strategies. Additionally, metabolic pathway and gene regulatory network topology analyses can identify key genes or competitive metabolic branches for targeted attenuation [78]. Flux balance analysis (FBA) is employed to simulate constrained metabolic fluxes, assessing the impact of enzyme gene attenuation on increasing the yield of target metabolites [79,80]. By integrating dynamic modeling, machine learning, and structural simulations, computational biology and modeling offer systematic methodologies for the design, optimization, and analysis of gene attenuation [81]. These approaches not only enhance research efficiency but also provide robust theoretical and practical frameworks for precise gene expression regulation and metabolic pathway optimization.

## 2. Applications of Gene Attenuation Across Various Biological Systems

Gene attenuation has broad applications, as it aims to modulate the characteristics of organisms or cells by partially reducing the expression or function of specific genes. Below are the main application systems of gene attenuation. The effects of gene weakening on different organisms are summarized in Table 2. Gene attenuation has diverse and extensive applications across different biological systems, playing a crucial role in both basic research and practical applications. It provides flexible gene regulation tools to study the partial effects of genes or develop new therapeutic approaches without eliminating gene function.

In plant systems, gene attenuation is primarily used for crop improvement, quality enhancement, and defense mechanism development. For example, by using RNAi technology to knock down the *OsPHP1* gene in rice, it was found that this attenuation enhanced rice yield under salt stress and drought conditions. Specifically, knocking down *OsPHP1* improved photosynthetic efficiency, increased tiller and panicle numbers, and enhanced seed setting rate and grain filling rate. Moreover, the attenuation of *OsPHP1* led to the upregulation of antioxidant and stress response genes, reducing ion toxicity and increasing the accumulation of osmotic substances, thereby improving plant growth and yield under stress conditions [82].

AsRNA technology was also used to knock down two key genes in Arabidopsis: cytosolic fructose-1,6-bisphosphatase (cFBPase) and sucrose phosphate synthase (SPS), and their effects on photosynthetic carbon metabolism were investigated. Reducing the expression of cFBPase led to the accumulation of phosphorylated intermediates and increased starch synthesis, while reducing SPS expression compensated for the balance of carbon allocation without causing the accumulation of phosphorylated intermediates or increased starch synthesis [83]. In *Arabidopsis thaliana*, Draborg et al. weakened the *AtF2KP* gene by asRNA technology, which encodes 6-phosphate fructose-2-kinase/fructose-2,6-bisphosphatase (F6P, 2-K/F26BPase). A part of the *AtF2KP* gene (930 bp fragment) was inserted into the control of the cauliflower mosaic virus 35S promoter in an antisense direction and introduced into Arabidopsis thaliana. Through this method, they successfully reduced the activity of F6P, 2-K/F26BPase, thereby reducing the level of fructose-2,6-diphosphate (Fru-2,6-P_2_). When the level of Fru-2,6-P_2_ decreased, the inhibition was weakened, resulting in an increase in sucrose synthesis and a decrease in starch synthesis, indicating that the ratio of sucrose to starch was 2 to 3 times higher than that of wild type [84].

Used asRNA technology to attenuate the expression of the *iaaM* and *ipt* genes in tobacco (Nicotiana tabacum). By constructing transgenic tobacco plants containing antisense copies of *iaaM* and *ipt*, the study found that these transgenic plants formed tumors with significant but incomplete gene expression suppression upon inoculation with the pathogenic Agrobacterium tumefaciens strain. These tumors exhibited weakened morphology and were capable of regenerating into complete plants. Northern blotting analysis revealed that the mRNA levels of the *iaaM* and *ipt* genes were reduced in the transgenic plants, indicating that the asRNA strategy effectively inhibited gene expression related to tumor formation, although full resistance to crown gall disease was not achieved [85]. By expressing asRNA complementary to the mRNA of the target gene, the expression level of the target gene can be reduced. For example, inhibition of polygalacturonase (PG) gene expression in tomato fruit by asRNA technology has been mentioned in the literature, thereby delaying fruit ripening and prolonging shelf life [86].

In summary, gene silencing technology can enhance plant production performance in multiple ways by precisely regulating gene expression, including improving stress resistance, regulating metabolism, and increasing yield and quality. Therefore, the application of gene silencing in plant systems has tremendous potential, especially in addressing environmental stress and improving crop traits.

In animal systems, gene attenuation technologies provide effective tools for gene function studies and transgenic animal production, offering major scientific and applied value. Gene attenuation, particularly RNAi technology, can effectively suppress the expression of specific genes, aiding in the study of gene functions. For example, RNAi targeting the myostatin (*MSTN*) gene successfully induced a double-muscle phenotype in zebrafish, providing valuable experimental data for understanding muscle development-related genes [87].

Gene attenuation can also be applied to produce transgenic animals with specific traits [88]. For example, RNAi technology has been used in goats, cows, and pigs to suppress the expression of specific genes, such as the *MSTN* gene, greatly improving meat quality [89]. Additionally, lentivirus-mediated RNAi can suppress the expression of porcine endogenous retroviruses (PERV), offering an effective strategy to reduce the risk of virus transmission in xenotransplantation [90].

By adding double-stranded RNA to the artificial diet, the RNAi response was successfully induced in the Western corn rootworm. In the experiment, dsRNA targeting multiple key genes (such as V-ATPase A, V-ATPase D, V-ATPase E, and α-tubulin) can dramatically reduce the mRNA levels of corresponding genes in pests, resulting in gene weakening. The larvae of western corn rootworm ingesting dsRNA showed obvious growth retardation. On the third day after ingestion of dsRNA, the weight of larvae was substantially lower than that of the control group. During the 12-day experimental period, dsRNA treatment for key genes substantially increased the mortality of pests. For example, dsRNA targeting the V-ATPase A gene can achieve a lethal concentration (LC_50_) of 50% at a concentration of 5.2 ng/cm^2^, indicating that RNAi can effectively inhibit pest growth and lead to death [91]. Unlike traditional gene knockout, gene attenuation allows partial suppression of gene expression without completely losing the function of the target gene, thus avoiding the adverse consequences, such as embryonic lethality, associated with gene knockout. This makes RNAi technology more widely applicable across multiple species. Overall, the application of gene weakening in animal systems has not only advanced the study of gene function but also provided important technological means for transgenic animal production, agricultural pest control, and the reduction in disease transmission.

In microbial systems, CRISPR technology has been used for gene attenuation to increase the yield of poly(3-hydroxybutyrate-co-4-hydroxybutyrate) [P(3HB-co-4HB)]. Specifically, by designing specific single-guide RNAs (sgRNAs), the CRISPR system was used to suppress the expression of key genes in competing metabolic pathways for 4-hydroxybutyrate (4HB) synthesis (Figure 2). For example, by attenuating genes involved in competing pathways (such as *sad*, *sdhA*, *sdhB*, *sucC*, and *sucD*), more metabolic flux was directed towards 4HB synthesis, increasing the 4HB content in P(3HB-co-4HB) from 1.4 mol% to 18.4 mol% [92].

This gene attenuation strategy not only increased the 4HB content but also enhanced the overall polyhydroxyalkanoates (PHA) production, achieving a 72% increase. Desai and Papoutsakis used asRNA technology to regulate the gene expression of *Clostridium acetobutylicum* and constructed different lengths of asRNA to down-regulate the expression of pho-sphotransbutyrylase, which reduced the concentration of acetone and butanol by 75% and 96%, respectively. Down-regulation of the expression of butyrate kinase increased the production of butanol and acetone by about 35% and 50%, respectively, indicating that the asRNA strategy to weaken specific genes can dramatically affect the metabolic pathway and product formation of *Clostridium* [93]. Wang et al. applied asRNA technology to modify the xylose metabolism pathway in filamentous fungi, effectively suppressing the expression of key enzymes (xylose dehydrogenase) involved in xylitol metabolism, achieving a 65% suppression rate, and increasing xylitol accumulation to 2.37 g/L, five times higher than in the initial strain. Extensive evidence demonstrates that asRNA technology can effectively suppress gene expression and even achieve functional gene knockout. Wang et al. conducted a systematic study on the effects of different gene attenuation strategies using various asRNA and GFP constructs. In large-scale screenings, they found that when asRNA targeted the *yngH* gene, it substantially affected the activity of the rate-limiting enzyme acetyl-CoA carboxylase (ACCase) in the surface-active compound synthesis process, leading to a significant reduction in surface-active compound production and enzyme activity compared to other experimental groups. Based on this finding, the research team applied a gene overexpression strategy, successfully increasing the surface-active compound yield in genetically modified strains to 13.37 g/L, a 43% increase compared to the control group [94]. Liu et al. enhanced the supply of NADPH by reducing the expression of glucose-6-phosphate isomerase (encoded by *pgi*). Five different asRNA fragments were designed to inhibit the expression of the *pgi* gene by asRNA technology. These asRNA fragments can bind to the mRNA of the *pgi* gene, thereby preventing its normal translation, thereby reducing the activity of the enzyme encoded by the *pgi* gene. The results showed that the recombinant strain carrying RNA fragment 5 produced 1.02 g/L 2-hydroxyadipic acid, which was 23.94% higher than that of the control strain without *pgi* weakening [95].

Yin et al. utilized automatically designed sRNAs to regulate the synthesis of the metabolite ethyl acetate, targeting the *ldh* and *pta* pathways that produce byproducts from the degradation branches. Specific sRNAs were designed and constructed to target these two genes in order to downregulate their byproducts. The results demonstrated that inhibition of *ldh*, *pta*, or both individually led to an increase in ethyl acetate titers. Additionally, inhibition of *pta* resulted in higher cell density and total yield, although it reduced the specific cell yield. This indicates that downregulation of *pta* not only promotes synthesis but also drives the TCA cycle, thereby enhancing biomass production [96]. Fu et al. successfully simulated the regulatory mechanism under nitrogen limitation conditions by downregulating the expression of NAD⁺-dependent isocitrate dehydrogenase (*IDH*) using CRISPRi and RNAi technologies, which enhanced the yield of itaconic acid (IA). In particular, the application of RNAi technology resulted in a 314% increase in IA yield, reaching 3.14 g/L [97].

**Table 2 microorganisms-13-00927-t002:** Application and strategy of gene weakening in different biological systems.

System	Organisms	Strategy	Application	Ref.
Plant System	Rice	RNAi	It can considerably enhance the salt and drought tolerance of rice, as well as increase its yield under stress conditions.	[82]
	*Arabidopsis thaliana*	asRNA	The reduction in SPS in Arabidopsis maintains carbon allocation balance.	[84]
	*Nicotiana tabacum*	asRNA	In tobacco plants, inhibiting the expression of tumor formation genes enhances disease resistance potential.	[85]
Animal System	Zebrafish	RNAi	Induces double-muscle phenotype in zebrafish, enhances muscle development.	[87]
	Insect pests	RNAi	Reduces expression of porcine endogenous retrovirus, decreasing virus transmission risk in xenotransplantation.	[91]
Microbial System	*E. coli*	CRISPRi	The content of 4HB further increased from 1.43 mol% to 18.4 mol%, an increase of about 12.8 times.	[92]
	*Clostridium acetobutylicum*	asRNA	The production of butanol and acetone was increased by about 35% and 50%, respectively, and the yield of lactic acid was effectively increased by the asRNA strategy.	[93]
	*E. coli*	asRNA	By using asRNA technology to weaken the *pgi* gene, the production of 2-hydroxyadipic acid was increased by about 24%, and the intracellular NADPH supply was enhanced.	[95]
	*E. coli*	sRNA	By weakening the translation of the *murE* gene by sRNA, the production of cadaverine could be increased to 12.6 g/L, which was 31% higher than that of the basic strain.	[98]

Shi et al. improved the production of 4-hydroxyisoleucine (4-HIL) by optimizing the RBS to enhance the synthesis of valine. It is known that the transcription factor *OdhI* inhibits the activity of ODHC, preventing the conversion of α-KG to downstream metabolites. Different strengths of RBS sequences (R8, R15, R5, R10, R9, R11) were used to regulate the expression of *odhI*, resulting in strains SF02-SF07. Compared to the control strain SF01, the engineered strains exhibited growth inhibition. Specifically, SF02, SF04, and SF06 showed reduced sugar consumption and lower 4-HIL production, with a large accumulation of glutamate, suggesting that weakening ODHC leads to a reduction in the TCA cycle, and more α-KG is diverted to glutamate production. SF03 and SF07 accumulated isoleucine while increasing 4-HIL production, indicating that using an appropriate RBS strength for gene expression helps to increase the yield [64] (Figure 2). Overall, gene attenuation techniques enable more efficient microbial production systems, reducing byproduct formation and enhancing product yields by precisely modulating gene expression, thus holding great potential for industrial biotechnology applications.

Moreover, Na et al. designed a series of specific sRNAs, which can accurately identify and bind to target mRNAs and effectively inhibit mRNA translation. In *E. coli* for cadaverine production, the yield of cadaverine was increased from 1.4 g/L to 2.15 g/L by weakening the translation of the *murE* gene. After further optimization, the yield of cadaverine was increased to 12.6 g/L by high-density fermentation, which was 31% higher than that of the basal strain [98]. In butanol production, RNAi technology was used to attenuate competing metabolic pathways, dramatically improving butanol yield and productivity by 5.4-fold and 3.2-fold, respectively, while reducing the generation of byproducts in the later stages of industrial production. Wu et al. proposed using CRISPRi technology for fine-tuning bacterial metabolic pathways to increase flavonoid production. By attenuating genes related to increased acetyl-CoA concentration (such as *ppsA*), they analyzed the impact of acetyl-CoA on cell growth. To maximize the regulatory effect, the researchers designed three sgRNAs with different inhibition efficiencies targeting different binding sites of the target genes. Based on a comprehensive analysis of the sgRNA inhibition effects and acetyl-CoA binding, they identified the optimal sgRNA design site, enabling the production of compounds derived from malonyl-CoA without adding substrates, thereby reducing industrial production costs. The CRISPRi-regulated cells reduced the flux of unnecessary byproducts and increased the flux of precursors and cofactors, optimizing the entire flavonoid production process and better coupling it with cell growth [99]. The study of Kim et al. proved the successful application of CRISPRi technology in *E. coli*, which can weaken the genes involved in the NADH consumption pathway and avoid the formation of byproducts. CRISPRi is used to weaken the competitive pathway and redistribute carbon flux to butanol production. These four genes are Pta (phosphate acetyltransferase), which is involved in acetic acid production; FrdA (fumarate reductase), which is involved in succinic acid production; LdhA (lactate dehydrogenase), which is involved in lactic acid production; and AdhE (aldehyde/alcohol dehydrogenase), which is involved in ethanol production. Under the condition of using glycerol as a carbon source and weakening the four genes *pta*, *frdA*, *ldhA,* and *adhE*, the yield of n-butanol increased by 2.1 times. The yield and productivity of n-butanol were increased by 5.4 times and 3.2 times, respectively, while the formation of byproducts, separation pressure and production cost were reduced [100].

Li et al. reduced the gene expression level by regulating the promoter of the *metK* gene. Specifically, the natural promoter of the *metK* gene in *E. coli* W3110 was replaced with the metK84 p promoter. MetK84 p is a mutant promoter, and the base at 150 changed from A to G, resulting in a significant decrease in the expression level of the *metK* gene, which successfully reduced the synthesis of SAM. This reduced the process of methionine being converted into SAM, so that more methionine could be accumulated in the medium, and the yield of methionine increased from 0 to 0.11 g/L [101]. These studies show that gene attenuation has major applications in metabolic engineering by precisely regulating key genes, optimizing metabolic fluxes, and improving the yield of target products.

The Leucine-Valine-Alanine (LVA) tag is a synthetic protein degradation tag designed to guide the rapid degradation of proteins with this tag by proteasomes in cells. Zheng et al. reduced the expression level of MS protein by adding LVA tags to the MS gene, thereby reducing the consumption of malonyl-CoA. By reducing the expression level of MS protein, the production of malic acid was reduced so that more acetyl-CoA and oxaloacetic acid could be used for the synthesis of malonyl-CoA. Finally, the malonyl-CoA level of the *E. coli* BL21.C33 strain was observably increased to 1.165 nmol/mg DCW, which was 26 times higher than that of the wild-type strain (0.045 nmol/mg DCW) [102]. By introducing RepA protein degradation tags to the 5′ end of the turboprop gene, the expression level of the TurboRFP protein was successfully reduced, and an efficient dual gene expression regulatory system (RR system) was constructed by binding to the double-tandem theophylline riboswitch (R2). This system not only substantially reduces the level of the target protein but also improves the flexibility and efficiency of gene regulation, providing new tools for the construction of complex gene circuits in synthetic biology [103].

In *E. coli*, a dynamic regulatory loop based on quorum sensing was used to attenuate the activity of the key enzyme ODHC (α-ketoglutarate dehydrogenase complex). This weakening is achieved by replacing the natural promoter of the ODHC gene with a quorum sensing-regulated PeaS promoter. In the absence of AHL (3-oxohexanoyl-homocysteine lactone), the transcription regulator EsaRI70 V binds to the PeaS promoter to trigger transcription; with the accumulation of AHL, the binding of EsaRI70V to the PeaS promoter was destroyed, thereby inhibiting the activity of the PeaS promoter and dynamically down-regulating the activity of ODHC. By dynamically weakening the activity of the ODHC gene, the yield of 4-Hyp was increased from 15.7 g/L to 54.8 g/L, which was increased by about 3.5 times. This metabolic flux redirection strategy based on dynamic regulation provides a new idea for metabolic engineering. By dynamically adjusting the activity of key enzymes, intracellular metabolic resources can be utilized more efficiently, and the yield of target products can be improved [104].

In plant, animal, and microbial systems, gene attenuation is a highly valuable tool. In plants, it helps to improve crop yield and stress resistance. In animals, it aids in gene function research and the creation of genetically modified animals. In microbial systems, it enhances product yield, optimizes metabolic pathways, and offers tremendous potential for biotechnology applications. The ability to precisely regulate gene expression without fully eliminating gene function makes gene attenuation a versatile and powerful strategy for various biological and industrial purposes. In summary, gene attenuation technology in metabolic engineering shows great potential in optimizing metabolic pathways, increasing target product yield, reducing byproduct formation, improving production efficiency, and lowering production costs. By accurately controlling the expression of key genes, researchers can effectively direct metabolic resources toward target products, providing more efficient and controllable production strategies for bio-manufacturing and industrial production. These advantages make gene attenuation an indispensable tool in metabolic engineering, laying the foundation for large-scale production of various industrial bioproducts.

## 3. Challenges and Discussion

Gene attenuation technology holds major potential for application in metabolic engineering, but it also faces several challenges that must be overcome to achieve effective use. These challenges include nonspecific effects, off-target effects, and metabolic bottlenecks, which can complicate the expected outcomes [105,106]. Gene attenuation technologies, such as RNAi and CRISPRi, are prone to nonspecific effects. These technologies may not be entirely specific to the target gene, potentially affecting the expression of off-target genes and leading to side effects or adverse reactions [107].

In RNAi, siRNAs often exhibit sequence similarity with non-target transcripts, leading to off-target effects. This may trigger immune responses or result in unintended gene silencing. Off-target effects based on RNAi can impact multiple unintended genes by mimicking the action of endogenous miRNAs, resulting in broad and often unpredictable changes in gene expression. In fact, miRNA-like off-target effects pose a serious challenge in large-scale RNAi screening approaches, where many positive results are false positives caused by such off-target activity [108,109]. Therefore, strategies to control, reduce, or even eliminate these off-target effects are urgently needed.

In RNAi-mediated pest control, these off-target effects may not be a major concern for plant systems, as such translational regulatory mechanisms are likely relatively limited. However, in strategies where plants express siRNA or shRNA that are ingested by specific animals and exert toxicity against them, off-target effects must be carefully considered. For instance, non-target animals may also ingest these RNAs, and even in the absence of fully complementary target RNAs, partial complementary sites could still influence gene expression through the endogenous miRNA system.

To mitigate these off-target effects, chemical modifications can be made in two ways. First, 2′-O-methylation can be performed on the seed region of the siRNA, particularly at the second position from the 5′ end [110,111]. This modification weakens the interaction between the guide strand and the target RNA, reducing miRNA-like off-target effects without compromising the silencing of the target gene. Second, locked nucleic acid (LNA) incorporation involves embedding LNA into the seed region, which enhances the specificity of the siRNA and reduces off-target effects. LNA is a chemically modified nucleotide that links the 2′-OH group to the 4′-carbon atom via a chemical bond, thereby increasing the stability and specificity of the siRNA [112]. Another strategy is siRNA pooling. SmartPools consist of four different siRNAs targeting different positions of the same target gene, each with a unique off-target signature [113]. By combining them, the concentration of each individual siRNA is reduced, thereby diminishing off-target effects. SiPOOLs are highly complex and well-defined siRNA pools that can contain up to 30 distinct siRNAs. These pools can effectively eliminate off-target effects, even when a single siRNA with pronounced off-target effects is included. Lastly, optimizing siRNA design is crucial. To avoid seed region matching, the design of siRNA should prevent complete matches between the seed region (positions 2–7 or 2–8) and the 3′ UTR of non-target genes. This can be achieved through computational tools and experimental validation. Thermodynamic parameter optimization involves fine-tuning the thermodynamic properties of siRNA to select highly functional duplexes, thereby reducing off-target effects.

Similarly, in the CRISPR-Cas9 system, when guide RNAs (gRNAs) direct the Cas9 nuclease to genomic loci with imperfect sequence matches, off-target effects can occur, leading to unintended DNA cleavage. These off-target mutations, including insertions, deletions, or larger structural changes, may pose risks for therapeutic applications [114]. To address off-target effects, engineered variants of dCas9 are under development. Recently, an expanded PAM SpCas9 variant, known as xCas9, has been established. xCas9 exhibits dramatically higher DNA specificity compared to the commonly used SpCas9, resulting in markedly reduced off-target effects at genome-wide target sites [115]. Secondly, the length of sgRNA is optimized, which is responsible for identifying the target DNA sequence for Cas9 cleavage, thereby improving specificity and reducing non-target DNA cleavage in the CRISPR-Cas9 gene editing system [116]. RNA-guided DNA cleavage domain (Cas9 nicking enzyme) linked to reverse transcriptase is also used and compounded with Prime editing guide RNA (pegRNA). Prime editing allows a small amount of base insertion, deletion, and base conversion in human cells [117]. In addition, a chemically induced anti-CRISPR protein (AcrIIA4) was designed to prevent Cas9 DNA binding, which considerably improved the specificity and safety of the CRISPR-Cas9 system. In mammalian cells, the use of anti-CRISPR proteins to interfere with Cas9 DNA binding activity considerably reduced non-target activity [118]. At the same time, studies have shown that the enhanced specific SpCas9 (eSpCas9) variant reduces the non-target effect while maintaining strong target cutting [119]. Mitigating off-target effects in RNAi and CRISPRi enhances the precision of these technologies, thereby improving gene attenuation efficiency. It also increases the reliability of experimental results by reducing false-positive rates. In agricultural applications, this leads to more effective control of target pests [120].

Gene attenuation technology significantly enhances the performance of industrial microbial strains by precisely regulating gene expression levels while minimizing genetic modification intensity. Through metabolic flux redirection using CRISPRi, the 4HB content in P(3HB-co-4HB) was increased.

Even with these hurdles, the prospects for gene attenuation technology in metabolic engineering remain hopeful. Upcoming studies aim to amalgamate gene attenuation techniques with various metabolic engineering approaches, including metabolic flux analysis, synthetic biology, and systems biology, to enhance the accuracy of metabolic regulation. Combining multiple technologies will address the present difficulties of gene attenuation methods, including unintended impacts and indiscriminate targeting. With the ongoing advancement of gene editing methods, scientists are crafting more precise and effective instruments to improve the accuracy of targeting gene attenuation. Enhancing CRISPR systems, for instance, to minimize unintended impacts and boost the identification of target genes, will increase the efficiency of gene attenuation.

Developing innovative delivery systems and carriers represents a vital research path. Scientists are concentrating on creating innovative nanoparticle carriers and delivery mechanisms to enhance the internal stability and efficiency of RNAi and CRISPR molecules. This approach will enhance the effectiveness of gene silencing across various organisms, thus broadening the scope of gene attenuation technology applications.

The use of computational models and systems biology techniques is on the rise for forecasting how gene attenuation affects metabolic networks. Such techniques will aid scientists in enhancing their comprehension and management of metabolic flow, refining metabolic engineering blueprints, and boosting the production of desired products. Methodical methods are capable of replicating various strategies for gene suppression and possible adaptive reactions, aiding in the creation of more efficient metabolic engineering tactics.

With the advancement of technology, the use of gene attenuation techniques in clinical and industrial biotechnological fields is set to broaden. This encompasses the application of gene attenuation techniques for the creation of novel medications, boosting the efficacy of vaccine production, or augmenting the efficiency of microbial production facilities to ensure sustainable bioproduction. Such technological progress will lead to effective and eco-friendly bioproduction methods, especially in the realms of pharmaceuticals and energy production. To sum up, despite the numerous difficulties gene attenuation technologies encounter in metabolic engineering, their extensive practical use cannot be ignored. Gene attenuation technologies, by tackling unintended consequences, enhancing the effectiveness of gene attenuation, and deepening insights into metabolic network intricacies, will lay the groundwork for wider use in metabolic engineering. As new tools, delivery systems, and computational techniques evolve, the significance of gene attenuation technology is set to rise in clinical and industrial sectors.

## 4. Conclusions

Gene attenuation, a critical technology in metabolic engineering, holds significant importance and future prospects in achieving precise regulation and efficient optimization of target product synthesis. By reducing the activity or expression levels of key enzymes in non-target pathways, gene attenuation minimizes metabolic flux diversion and byproduct accumulation, thereby enabling efficient reconstruction of metabolic pathways and dramatically enhancing the yield of target metabolites. In the future, gene attenuation is anticipated to benefit from advances in synthetic biology tools, such as CRISPRi technology, RNAi, and programmable regulatory elements, facilitating its broader application across diverse biological systems. The widespread application of gene attenuation is expected to improve the biosynthetic efficiency of target products and promote sustainable development in energy, pharmaceuticals, agriculture, and environmental sectors. Its significance lies in providing robust technical support for green biomanufacturing and resource-efficient utilization, further solidifying the role of metabolic engineering as a cornerstone in the future bioeconomy.

## Figures and Tables

**Figure 1 microorganisms-13-00927-f001:**
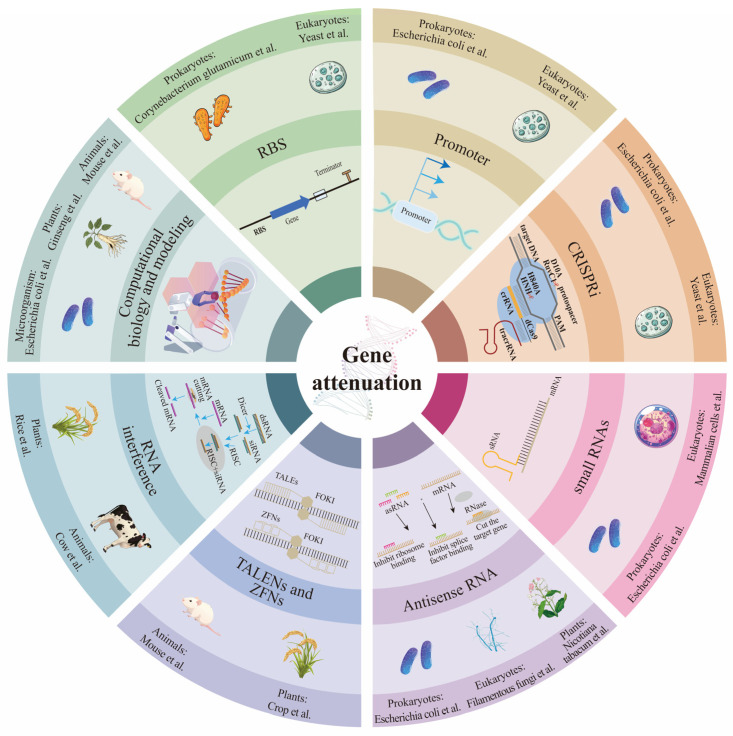
Mechanisms of gene weakening and commonly used organisms. AsRNA, sRNAs, RNAi, CRISPRi technology, regulation of promoters and ribosome binding sites (RBS), TALENs, ZFNs, and computational biology and modeling can achieve precise control of target gene expression levels. The arrows in RNAi and asRNA represent the order of each process in the pathway; the deep to shallow arrows in the promoter indicate that the promoter strength is from strong to weak.

**Figure 2 microorganisms-13-00927-f002:**
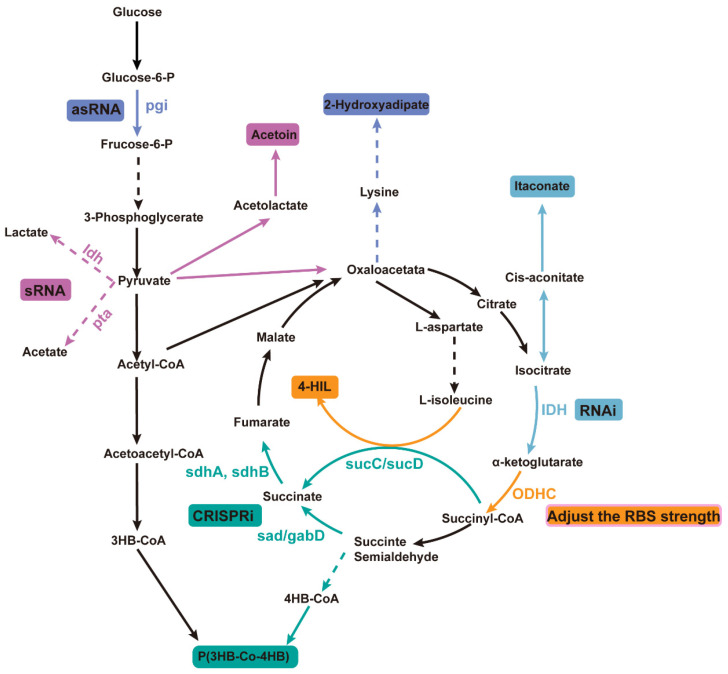
Improving product yield by utilizing different weakening mechanisms in microorganisms. Different colors represent different products using different weakening methods. Automatic design of sRNA was used to downregulate *ldh* and *pta*, promoting the synthesis of the metabolite ethyl acetoin; an AsRNA strategy was used to downregulate the expression of glucose-6-phosphate isomerase (encoded by *pgi*) to promote the production of 2-hydroxyhexanedioic acid; RNAi technology was used to downregulate the expression of NAD⁺-dependent isocitrate dehydrogenase (IDH), thereby increasing the yield of itaconic acid (IA); RBS strength adjustment was used to inhibit the activity of ODHC, leading to an increase in the production of 4-HIL; CRISPRi technology was employed to weaken genes involved in competing pathways to improve the yield of P(3HB-co-4HB). The dotted line arrow indicates that there is a multi-step reaction, and the real line arrow indicates a one-step reaction; the color consistency of the product and the attenuation technology indicates that the product is used to achieve the purpose of gene weakening.

**Table 1 microorganisms-13-00927-t001:** A summary of the main differences in gene regulation.

Strategy	Description	Methods	Applications	Ref.
Gene attenuation	Reduces gene expression or lowers product activity to weaken function.	RNAi, CRISPRi, etc.	Studying the effects of reduced gene expression on an organism.	[30,31]
Gene knockout	Completely removes or deactivates a gene.	CRISPR-Cas9, homologous recombination.	Investigating a gene’s function in biological processes.	[32]
Gene overexpression	Increases gene expression to enhance product levels.	Promoters, introducing extra copies.	Boosting the synthesis of target products.	[33]

## Data Availability

No new data were created or analyzed in this research.

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
