# Peer review of "The Strategy and Application of Gene Attenuation in Metabolic Engineering"

_microorganisms, 2025, doi:10.3390/microorganisms13040927_

Round 1

Reviewer 1 Report

Comments and Suggestions for Authors

A review of the manuscript entitled "The strategy and application of gene attenuation in metabolic engineering." The manuscript provides a comprehensive overview of methodologies related to the attenuation of gene expression (gene attenuation) in metabolic engineering. The manuscript's subject matter is particularly relevant in the current research climate, as it aligns with prevailing trends in metabolic pathway optimization to enhance the efficiency of biotechnology production. A detailed review of various gene attenuation techniques was presented, accompanied by precise descriptions of their respective mechanisms of action. The study is noteworthy for its comprehensive and meticulous examination of the technologies employed. The presentation of information is meticulously organized, with clear structuring, and the integration of visual aids, such as Figure 1 and tabular representations (Tables 1 and 2), significantly enhances the clarity and comprehension of the material. However, it should be noted that several aspects could be refined or described in more detail.
The discussion of technological limitations and challenges could be expanded to include specific examples from the literature that would better illustrate the difficulties of practical implementation of the methods presented. Such examples could include modification stability, scaling issues, or precision of expression control in industrial settings. The potential risk of off-target effects, which is essential, especially for systems such as RNAi or CRISPR, should be discussed more extensively. It would be helpful to point out methods to minimize these effects. The section on prospects should be enriched with a more detailed analysis of the development of gene editing technologies, considering the latest developments and trends that may affect the development of the gene expression attenuation strategies discussed.
Furthermore, it would be beneficial to include case studies of practical applications successfully implemented on an industrial scale. This would reinforce the viability of the strategies outlined.

Author Response

Comments 1: The discussion of technological limitations and challenges could be expanded to include specific examples from the literature that would better illustrate the difficulties of practical implementation of the methods presented. Such examples could include modification stability, scaling issues, or precision of expression control in industrial settings.
Response 1: We sincerely appreciate your valuable feedback on this matter. We have accordingly made the revisions. In the revised draft, there is an addition regarding the precision of gene expression between lines 594 and 600, which may potentially affect non-target organisms.
Comments 2: The potential risk of off-target effects, which is essential, especially for systems such as RNAi or CRISPR, should be discussed more extensively. It would be helpful to point out methods to minimize these effects.
Response 2: We are grateful for your pointing out this issue. We have accordingly made the revisions. In the revised draft, between lines 590 and 644, there are additions regarding the off-target effects of RNAi and CRISPR, as well as solutions to these issues.
Comments 3: The section on prospects should be enriched with a more detailed analysis of the development of gene editing technologies, considering the latest developments and trends that may affect the development of the gene expression attenuation strategies discussed.
Response 3: Thank you for pointing out this issue. We agree with this comment. The development history of gene editing has been added to Lines 52-77 in the revised manuscript.
Comments 4: Furthermore, it would be beneficial to include case studies of practical applications successfully implemented on an industrial scale. This would reinforce the viability of the strategies outlined.
Response 4: We appreciate you pointing out this issue. We agree with this comment. In the revised draft, there is an introduction between lines 640 and 648.

Reviewer 2 Report

Comments and Suggestions for Authors

Zhang et al. review the use and impact of gene attenuation for metabolic engineering. Attenuation is a milder form of engineering compared with standard knock-out strategies and can be implemented to fine-tune expression as detailed in the review. The review shows the diverse approaches for attenuation, the applicability to different host systems and constructive examples. The content is suitable and the text is overall well written and formatted. In general, the text could become more accessible when including more visualizations and tables.

Suggestions for revision:

  1. Figure 1 could contain information which strategy is usable in which host system (prokaryote, eukaryotic)
  2. Organism names and genes should be italics and there are numerous occasions when this needs correction.
  3. L26: Metabolic engineering is about 'controlling' 'cellular activities', but not improving. 'Improving' would fit if a biotechnological objective is considered.
  4. L147: '.RNA.' seems lost.
  5. The references need a closer look, e.g., #1 not every letter of the surname needs to be in capital, #23-#28 the author names need to be properly given
Comments on the Quality of English Language
  1. Some paragraphs are too long. E.g., L142-182 could probably be broken with a new paragraph starting at 168. Other paragraphs are L328-368, L369-397, L398-460, L503-538.
  2. The hyphenation is sometimes wrong and often inaccurate. L30 microor-ganism>micro-organism, L57 bal-ance>balance, L59 overex-pression>over-expression, L347 reduc-ing>re-ducing, L462 Microorgan-isms>Micro-organisms
  3. The term 'significant' is used too indiscriminantely. There are 20 positions, many of them should be reformulated to avoid repetition and false word usage.
  4. L462: Not every word in the first caption sentence need to be capitalized.
  5. L105: Write out abbreviation at its first use, and keep the first letter of the abbreviation small as in L110

Author Response

Comments 1: Figure 1 could contain information which strategy is usable in which host system (prokaryote, eukaryotic).
Response 1: We truly appreciate you pointing out this issue. We agree with this comment and have accordingly included the commonly used host information in Figure 1 of the revised manuscript.
Comments 2: Organism names and genes should be italics and there are numerous occasions when this needs correction.
Response 2: We sincerely thank you for pointing out this problem. We have accordingly made the revisions.
Comments 3: L26: Metabolic engineering is about 'controlling' 'cellular activities', but not improving. 'Improving' would fit if a biotechnological objective is considered.
Response 3: Thank you for pointing this out. We agree with this comment. Therefore, we have replaced "improving" with "controlling" in line 27 of the revised manuscript.
Comments 4: L147: '. RNA.' seems lost.
Response 4: Thank you for emphasizing this. We agree with this opinion and accordingly implement the proposed changes in the revised manuscript L177.
Comments 5: The references need a closer look, e.g., #1 not every letter of the surname needs to be in capital, #23-#28 the author names need to be properly given
Response 5: Thank you for highlighting this point. We have accordingly made the revisions.

Comments on the Quality of English Language.
Comments 1: Some paragraphs are too long. E.g., L142-182 could probably be broken with a new paragraph starting at 168. Other paragraphs are L328-368, L369-397, L398-460, L503-538.
Response 1: We sincerely thank you for pointing out this problem. We have accordingly made the revisions.
Comments 2: The hyphenation is sometimes wrong and often inaccurate. L30 microor-ganism>micro-organism, L57 bal-ance>balance, L59 overex-pression>over-expression, L347 reduc-ing>re-ducing, L462 Microorgan-isms>Micro-organisms.
Response 2: We are very grateful for the issue you have raised. We have accordingly made the revisions.
Comments 3: The term 'significant' is used too indiscriminantely. There are 20 positions, many of them should be reformulated to avoid repetition and false word usage.
Response 3: Thank you for your questions. We have accordingly made the revisions.
Comments 4: L462: Not every word in the first caption sentence need to be capitalized.
Response 4: We thank you very much for pointing out this problem. Accordingly, we modified the manuscript L483 to emphasize this point.
Comments 5: L105: Write out abbreviation at its first use, and keep the first letter of the abbreviation small as in L110.
Response 5: Thank you for pointing this out. We agree with this opinion and modify it in manuscript L137.

Round 2

Reviewer 2 Report

Comments and Suggestions for Authors

Thank you very much for the final polishing of the manuscript.